ⓐ | **Open Peer Review** | Microbial Ecology | Research Article

# Biogeographical and phylogenetic constraints on horizontal gene transfer and genome evolution in *Streptomyces*

**Janani Hariharan,**[1] **Cheryl P. Andam,**[1] **Daniel H. Buckley**[1,2]

**ABSTRACT** The role of horizontal gene transfer (HGT) in shaping bacterial genomes is well recognized, but constraints on gene exchange and the degree to which these constraints shape genome evolution remain poorly described. In this study, we sought to determine whether geographic and phylogenetic distance constrains HGT within and between bacterial species. To address this question, we isolated strains ($n = 17$) of two closely related bacterial species, *Streptomyces griseus* and *Streptomyces pratensis* from two ecologically similar sites. We identified homologous recombination events within the core genomes of these species (557 recent and 457 ancient) and determined that patterns of recombination were constrained primarily by phylogeny rather than geography. Notably, shell accessory genes were over three times more likely to be shared between the same species than with non-related geographical neighbors. The richness of secondary metabolite gene clusters is highly variable with an average of 35–55 clusters per genome, depending on clade membership. The majority of secondary metabolite gene clusters (60%) are found in all strains, indicating that they were present in the most recent common ancestor of *S. griseus* and *pratensis*. We conclude that most HGT in the core and accessory genome is phylogenetically constrained, while HGT of shell genes is more likely influenced by geography. This outcome indicates that the predominant mechanisms of HGT favor high phylogenetic relatedness, and that rapid gene acquisition and loss in the accessory genome could aid with adaptation to contemporary environmental conditions.

**IMPORTANCE** Horizontal gene transfer (HGT) is a vital ecological and evolutionary force in microbiology, but we still lack a precise understanding of how precisely HGT acts on the gene pool of a species or genus. While HGT can complicate phylogenetic analyses and assumptions of homology, its role in adaptation and acquiring secondary metabolites should not be overlooked. Microbial ecologists agree that the pangenome is a shifting collection of genes that can be influenced by both vertical inheritance and ecological factors. This study examines how the *Streptomyces* pangenome is shaped by these two forces and offers an important quantitative insight into how HGT shapes bacterial genome dynamics.

**KEYWORDS** *Streptomyces*, horizontal gene transfer, biogeography, pangenome, comparative genomics, secondary metabolite, actinophage

**Peer Reviewer** Sébastien Bontemps-Gallo, Centre d'Infection et d'Immunite de Lille, Lille, France

Address correspondence to Janani Hariharan, jh069@bucknell.edu.

The authors declare no conflict of interest.

See the funding table on p. 15.

Horizontal gene transfer (HGT) is a strong evolutionary force that shapes the structure and dynamics of prokaryotic genomes. HGT can introduce genes with adaptive potential (1). For example, most metabolic innovations in the *Escherichia coli* genome were acquired through HGT (2). Traits associated with human and ecosystem health, such as antibiotic resistance and agricultural intensification, are mediated by HGT in natural and managed environments (3–6).

10.1128/spectrum.02958-25  **1**

Horizontally acquired DNA can be introduced into a recipient cell through conjugation, transformation, or transduction. Persistence of acquired DNA, however, requires insertion and integration into the recipient's genome. DNA can be integrated into the genome via homologous recombination (HR) as a result of *recA*-mediated repair, or by one of several mechanisms of non-homologous recombination (NHR). HR favors HGT between close relatives, as observed in the increase in frequency of HR in proportion to the sequence similarity of donor and recipient (7), while NHR proceeds via non-homologous end joining, which does not require sequence similarity between donor and recipient.

Phylogenetic affinity has consistently been identified as a driver of HGT patterns in bacteria. Plasmids and viruses can have narrow ranges of host specificity (8, 9), and restriction modification systems protect their host cells against infection by foreign DNA by un-methylating and subsequently cutting or destroying it (10). However, the amount of HR within a species also appears to vary by lifestyle. Obligate and opportunistic pathogens have some of the highest recombination rates, followed by free-living species. Endosymbionts and intracellular pathogens showed the lowest recombination rates (1). This variation suggests that the environment might also play an important role in HGT.

HGT events that are predicated on geographical proximity may even operate on a faster tempo; for example, Boucher and coworkers found that global *Vibrio* populations possessed a vertically inherited gene pool as well as a local gene pool that is rapidly transferred across species boundaries (11). Analysis of other microbial groups (including *Vibrio*, *Sulfolobus*, and *Prochlorococcus*) indicates that closely related species in different locations share little to no recent gene flow, leading to the formation of distinct population clusters (12). Thus, phylogenetic and geographical proximity both impact gene flow between populations, but the relative contributions of these variables to HGT are yet unknown for many microbial groups, and particularly for free-living terrestrial organisms.

In this study, we use a unique experimental design to investigate the quantitative contributions of phylogeny and geography towards HGT in two phylogroups of the bacterial genus *Streptomyces, S. pratensis*, and *S. griseus*. The genus *Streptomyces* (phylum Actinomycetota) is a highly diverse group found predominantly in soil. They are a major source of naturally derived antimicrobials and numerous bioactive, pharmaceutically relevant metabolites (13). While the clinical and agricultural importance of this genus has long been recognized, the factors driving its evolutionary history, ecology, and diversity remain poorly investigated.

Previous studies have reported extensive HGT within this genus (14–16). Genetic exchange in *Streptomyces* appears to be non-random with inter-species HGT occurring at a lower rate than within-species HGT (15). This suggests that there is a mechanism for constraining inter-species HGT. Other studies have found that habitat barriers limit homologous recombination rates within strains of *Streptomyces albidoflavus* (17). The quantitative contributions of phylogeny and geography toward generating patterns of diversity in this genus are, however, unclear.

The co-occurrence of distinct *Streptomyces* strains in nature, despite widespread HGT, presents a unique opportunity to investigate the dynamics of inter- and intra-species interactions at the genetic level. Strains in this study are evenly distributed across two sites (Washington and Wisconsin, USA) and two phylogroups (*S. griseus* and *pratensis*). This design allows us to contrast the frequency of gene transfer events and the magnitude of gene flux between co-localized and geographically distant strains. We hypothesize that (i) phylogenetic relatedness will dictate the core genome of each phylogroup, but (ii) geography is likely to play a role in the acquisition of unique genes that may have adaptive significance to the local habitat.

## MATERIALS AND METHODS

### Bacterial isolates and DNA extraction

Strains of *S. pratensis* (*n* = 9) and *S. griseus* (*n* = 8) were isolated from grass rhizosphere soil in sites from both Rhinelander, Wisconsin (RH; 45.63°N, 89.41°W) and Bothell, Washington (WA; 47.76°N; 122.21°W) (straight line distance = 2,493 km). These strains were part of an existing culture collection containing more than 1,000 *Streptomyces* from 15 sites across the United States (18). The sites are similar in their environmental characteristics: both sites are dominated by perennial grasses and have slightly acidic pH (Bothell: 5.3; Rhinelander: 5.9). Bothell has a summer subtropical climate with an average annual temperature of 11.6°C, annual precipitation of 83.1 cm, and soils that are loamy-skeletal inceptisols. Rhinelander has a warm summer continental climate with an average annual temperature of 5.6°C, annual precipitation of 85.31 cm, and soils that are sandy-loam spodosols. Soil was collected at 0–5 cm depth and air-dried, following which *Streptomyces* were isolated on glycerol-arginine media, pH 8.7, containing cycloheximide (300 mg/L) and rose bengal (35 mg/L), as previously described (19). The genetic diversity of isolates in this collection was initially assessed using partial *rpoB* sequences. Sub-groups of *S. pratensis* (pr) and *S. griseus* (gr) were named RH-PR, RH-GR, WA-PR, and RH-GR to indicate strains from each species isolated from each of the two sites (Rhinelander, Wisconsin and Bothell, Washington, respectively). DNA was extracted from purified cultures, which were grown by shaking at 30°C in liquid yeast extract-malt extract medium (YEME) containing 0.5% glycine (20), by using a standard phenol/chloroform/isoamyl alcohol protocol.

### Whole genome sequencing, annotation, and assembly

Genomic DNA over 10 kb was size selected using gel purification (MO-BIO, Carlsbad, CA) and submitted to the Cornell Biotechnology Resource Center for DNA sequencing. DNA libraries were prepared using the Nextera library preparation kit (Illumina Inc.), and all libraries were run together in multiplex using GS FLX Titanium series reagents on a GS FLX instrument. The contiguous sequences were assembled *de novo* using the A5 genome assembly pipeline (21). Annotations of the draft genomes were done using Prokka v1.14.6 (22). Assembly statistics for each genome are found in Table S1. To measure the genomic relatedness between strain pairs, we calculated the average nucleotide identity (ANI) values using fastANI v1.32 (23) with default parameters. Genomes with ANI values of 95% or higher were considered to be the same species, and partial *rpoB* sequence similarity was used to resolve species classification in cases where the ANI was close to but not exactly 95%.

Pangenome gene content was analyzed using the Roary pipeline (v3.13) (24). Orthologous groups of open reading frames (ORFs) from all strains were identified and clustered using OrthoMCL (25) as implemented in GET_HOMOLOGUES REF using default settings.

### Determination of HGT events

Draft genomes were aligned using Mugsy v1.2.3 (26), and a genome-based phylogeny was reconstructed using the maximum likelihood method as implemented in RAxML v8 (27). Effective recombination rates within populations were calculated using ClonalFrameML v1.11 (28).

All core genes identified by OrthoMCL were aligned with MUSCLE v3.8 (29) and concatenated using FASconCAT-G (https://github.com/PatrickKueck/FASconCAT-G). This alignment was used as the input to the fastGEAR algorithm (30), which detects recombination events. Selection pressure on recombination sites was assessed by calculating the ratio of non-synonymous substitutions to synonymous mutations (dN/dS) using PAML v4.8 (31). The Panther database (32) was used to obtain gene ontologies for genes shared or lost between groups of interest. For the Panther ontology analysis, only genes that have previously been identified in the *Streptomyces coelicolor* reference

genome are reported in text unless otherwise specified. HGTector (33) was used to detect potential external donors.

## Gene gain and loss estimation

Orthogroups were detected using the OrthoFinder pipeline (v2.3.8) (34) and subsequently used for gene gain and loss analysis using Notung v2.9.1 (35). Notung provided a count of genes gained, lost, and horizontally transferred at each node in the phylogeny. Genes gained or lost for specific groups of interest were functionally annotated using the COG database (36) via the eggNOG mapper (37). Annotations for orthogroups under selection were cross-verified using the eggNOG database in addition to the prokka annotations.

## Identification of secondary metabolite gene clusters (SMGCs)

Secondary metabolite biosynthetic gene clusters were identified using antiSMASH v3.0 (38). Codon-aware alignments were generated from the antiSMASH-identified nucleotide sequences using the pal2nal pipeline (39). These alignments were subsequently processed through the PAML pipeline to estimate the ratio of synonymous and non-synonymous substitutions ($\omega$) within each cluster to identify whether neutral, positive, or negative selection was acting on secondary metabolite gene clusters. Phylogenetic incongruence in tree topologies was evaluated using the Shimodaira-Hasegawa test (40) with 10,000 permutations as implemented in the R package phangorn (41). This test operates on the null hypothesis that all candidate trees have the same expected test statistic, and deviations from this assumption may indicate phylogenetic incongruence in the set of trees being tested.

## Identification of mobile genetic elements

Mobile genetic elements (MGEs) were identified relative to reference databases: aclame for plasmid and phage elements (42), ICEberg 2.0 for integrative and conjugative elements (43), and ISEScan for Insertion Sequence elements (44). Sequenced phage genomes with a host range specific to the *Streptomyces* genus from the Actinobacteriophage Database (45) were aligned with our draft genomes to identify actinophage signatures indicative of transductive gene transfer events.

## Statistical analyses

All statistical tests and data visualizations were performed in R 3.6.1 (R Core Team 2021). Results were considered significant when $P < 0.05$ at least. Specific packages and functions are indicated in the appropriate Results sections.

## RESULTS

### Sub-clades within species are consistent with geography

Phylogenetic reconstruction using shared orthologous core genes shows that subclades within each species are delineated by geography (Fig. 1). The average ANI between *S. pratensis* subclades WA-PR and RH-PR was 93.89 ± 0.05, and 95.96 ± 1.17 between *S. griseus* subclades WA-GR and RH-GR. The strain RH34 is divergent from other RH-GR strains (Fig. 1) but has higher ANI with these strains than to strains from the WA-GR subclade (Table S2). Intra-clade ANI was typically >99% for both species with the exception of RH34. The ANI between the two species groups was 84.9 ± 1.06, confirming their divergence according to currently accepted ANI metrics (23).

The *S. pratensis* strains have an average ANI of 96.72 ± 2.72% to the *S. pratensis* type strain and average ANI of 96.3 ± 2.9 with each other, indicating that they are all members of the *pratensis* phylogroup (Table S2). *S. pratensis* genomes contained 6,958 ± 271 genes, but genome size differed between subclades (8.18 Mb ± 0.13 for WA-PR, and 7.62 Mb ± 0.17 for RH-PR; *t*-test, $P < 0.001$). The *S. griseus* strains have 93.11 ± 0.12% ANI to the type strain for *S. griseus* and average ANI of 96.9 ± 2.1% with each other, indicating

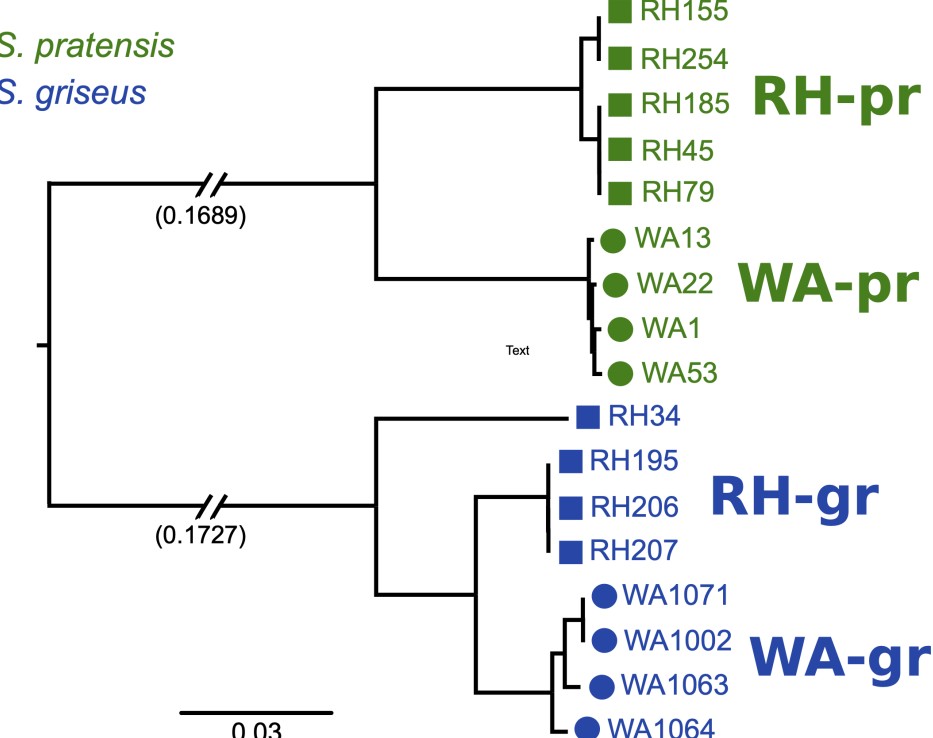

**FIG 1**   A maximum likelihood-based reconstruction of the core gene phylogeny shows strain differentiation based on their species grouping (*griseus* or *pratensis*) and site of isolation (Washington [WA] or Wisconsin [RH]).

that they are all members of a single species that is most closely related to *S. griseus*. *S. griseus* genomes contained 7,728 ± 219 genes, and genome size did not differ between subclades (8.74 Mb ± 0.92 for WA-GR, and 8.53 Mb ± 0.20 for RH-GR; *t*-test, *P* > 0.1). When comparing all 17 strains, genome size differed by both phylogeny and geography (two-way ANOVA, *P* < 0.001) and the interaction between phylogeny and geography, i.e., clade membership in Fig. 1 (two-way ANOVA, *P* < 0.05), which could be a result of differing ancestral gene content within each species as well as demographic events local to each habitat.

## Pangenome composition indicates genomic differentiation with respect to geography

We evaluated the pangenome composition for strains of *S. pratensis* and *S. griseus* by using the R package micropan (46). *S. pratensis* had a core genome consisting of 3,353 genes, while the *S. griseus* core genome had 3,269 genes. The core genome of *S. pratensis* had 1,765 genes not found in *S. griseus*, while the core genome of *S. griseus* had 1,601 genes not found in *S. pratensis*. The core genomes of the two species shared 491 genes.

Genomes of the same species typically exhibit a characteristic U-shaped gene frequency distribution (47, 48), in which most genes are found as either core or strain-specific genes, while intermediate frequency shell genes (those present in two or more strains but not present in the core) are relatively rare. The gene frequency distributions for both *S. pratensis* and *S. griseus* show an elevated number of intermediate frequency genes (Fig. 2), with a local maximum for genes that occur in 3–5 genomes. This pattern of gene frequency indicates intraspecies genetic structure that corresponds to geography.

Shell gene distribution was strongly influenced by phylogeny (Welch's *t*-test, *P* < 0.001). However, more shell genes were shared between different phylogroups at the same site (e.g., RH-PR and RH-GR) than between strains from different sites (e.g., RH-PR and WA-PR) (Fig. 3). RH34 shares relatively few shell genes with other strains of the

RH-GR subclade, and it also has an unusually high number of unique genes (3,611 unique genes) as compared with other strains (201 ± 188 unique genes). The number of unique genes in each genome varied by geography (WA > RH; 460 and 336, respectively) but not phylogeny (GR > PR; 269 and 148, respectively, excluding RH34) based on a two-way ANOVA ($P < 0.01$). The ANOVA result did not change when RH34 was excluded from the analysis.

## Core gene HGT is influenced by both phylogeny and geography

ClonalFrameML estimates high recombination rates in both *S. pratensis* and *S. griseus* (r/m of 8.1 and 6.3, respectively). For comparison, r/m values of other soil bacteria are generally in the range of 0.1–5.5 (49). *Streptomyces* have previously been observed to have unusually high r/m values that indicate high rates of gene exchange and recombination (15).

We identified 1,014 recombination events (557 recent and 457 ancestral) among shared orthologous genes using fastGEAR. Analysis of polymorphic sites by fastGEAR identified three main lineages comprised of WA-PR, RH-PR, and *S. griseus* (WA-GR + RH GR; Fig. 4A). Few recombination events were detected from *S. griseus* to *S. pratensis* strains ($n = 11$ recent and 0 ancient), and so we focus mainly on recombination events originating from *S. pratensis* subclades. Ancestral recombination events were only detected between the two *pratensis* clades, as all *griseus* strains clustered into a single fastGEAR lineage. The number of inter-species recombination events did not differ significantly by geography (one sample *t*-test, $P > 0.05$), and intra-species recombination events were significantly more likely than inter-species recombination (one sample *t*-test, $P < 0.001$). Given the small sample size of this study, we also estimated effect sizes of phylogeny and geography on the number of recombinations using Cohen's D as applied in the R package "effectsizes" (phylogeny = 1.34, geography = 0.51) (50). These results indicate that the extent of recombination is greater within species than between species regardless of site for the core genome.

A large number of recombination events were observed to originate from donors other than *Streptomyces* (Fig. 4B). External recombination events were significantly more common in genomes from WA (192 ± 23.9) than in genomes from RH (161 ± 14.5)

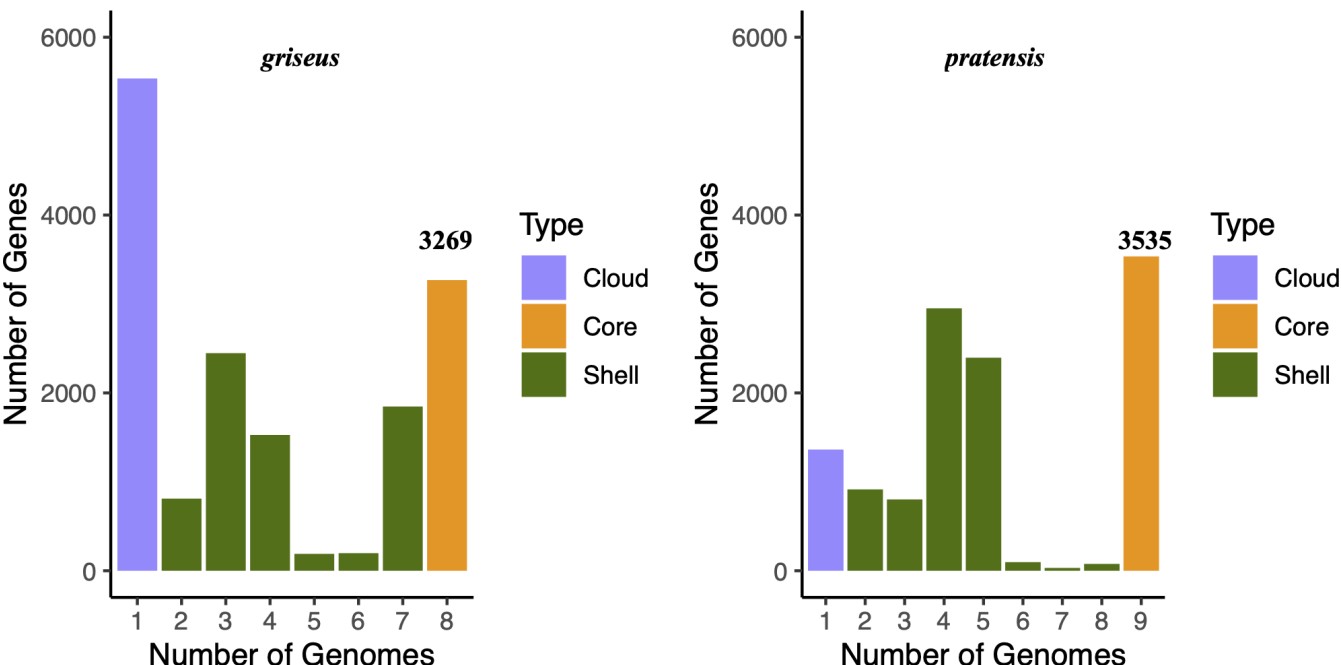

**FIG 2** Both *griseus* and *pratensis* strains show a large number of intermediate frequency shell genes that correspond with the geographical distribution of their subclades as depicted in Fig. 1.

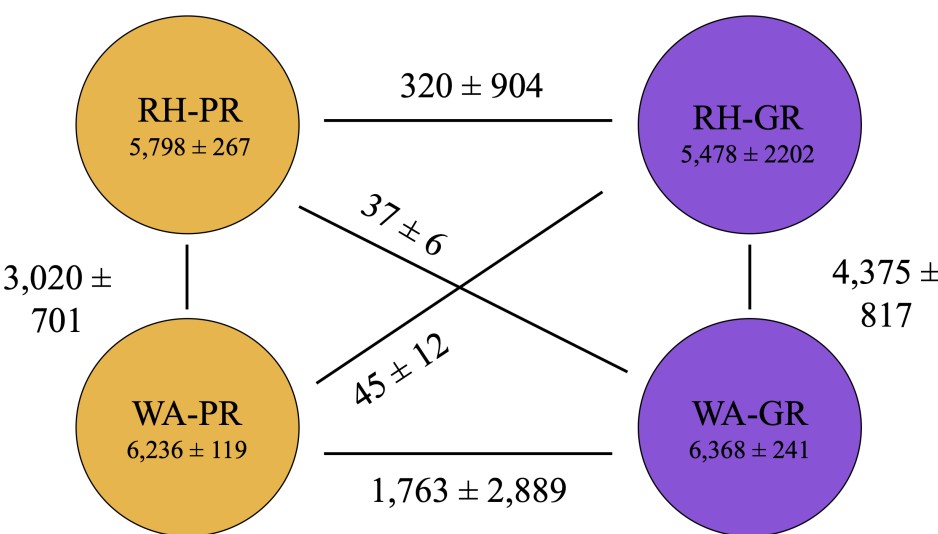

**FIG 3** Shell gene distribution across the *Streptomyces* strains is influenced by their phylogenetic and geographic relationships. The yellow and purple circles represent *pratensis* and *griseus* genomes, respectively. Most shell genes are shared within strains of the same species regardless of site. Of the shell genes shared between different species, significantly more are shared within site than across site (Student's *t*-test, RH34 excluded, *P* < 0.005).

(Welch's *t*-test, *P* < 0.005) as detected by the HGTector algorithm. Most external transfers came from within the class Actinomycetes (*n* = 2,686/2,848 or 94.3%).

## Shell gene evolutionary dynamics

Gene gain, loss, and duplication events were mapped to the reconstructed phylogeny as shown in Fig. 5. The number of genes lost in each strain is impacted by the interaction between phylogeny and site (*P* < 0.01, two-way ANOVA), with RH-GR strains having the lowest number of average gene losses (1,243 ± 452). Gene duplications are driven by site of isolation (*P* ≤ 0.001, Welch's *t*-test), with WA strains having more paralogs (31 ± 14) than RH strains (3 ± 2). RH34 was excluded from these calculations given that its genome dynamics are markedly different from those of its neighbors, and its unique dynamics are described later in this section.

A KEGG pathway analysis of gene loss shows clade-specific losses of aminoacyl-tRNA synthetases. WA-PR strains lost genes *glyS1* or glycyl-tRNA synthetase, and *pheS* or phenylalanyl-tRNA synthetase. The WA-GR clade lost genes encoding genes *tyrS*, *lysK*, *lysS* (tyrosyl-tRNA synthetase, lysyl-tRNA synthetases), and *leuB* involved in leucine biosynthesis. In Wisconsin, the RH-PR strains lost aminoacyl-tRNA synthetase genes *trpS*, *alaS*, *metG*, and *serS* (tryptophanyl-tRNA synthetase, alanyl-tRNA synthetase, methionyl-tRNA synthetase, seryl-tRNA synthetase), while the RH-GR strains did not show any losses associated with aminoacyl-tRNA biosynthesis genes. All clades lost genes involved in biosynthesis of secondary metabolites (streptomycin, neomycin/kanamycin/gentamicin, prodigiosin, terpenoid backbone) and genes required for vitamin B (thiamine and biotin) metabolism, and WA-GR lost genes encoding streptomycin and chloramphenicol resistance.

Interestingly, the loss of amino acid biosynthesis capability was accompanied by the gain of amino acid synthesis genes or gene clusters in some cases. WA-PR strains gained genes to encode tryptophanyl-tRNA synthetase, WA-GR gained a component of phenylalanyl-tRNA synthetase, and RH-GR gained *serS* (seryl-tRNA synthetase), *korB*, *pdhC*, and *leuC* (genes involved in leucine and isoleucine biosynthesis). All Wisconsin strains gained genes for asparagine synthesis (*asnB*).

Gene duplication events are lower by an order of magnitude compared to loss (Table S3), but duplications are of particular importance in identifying genes under positive

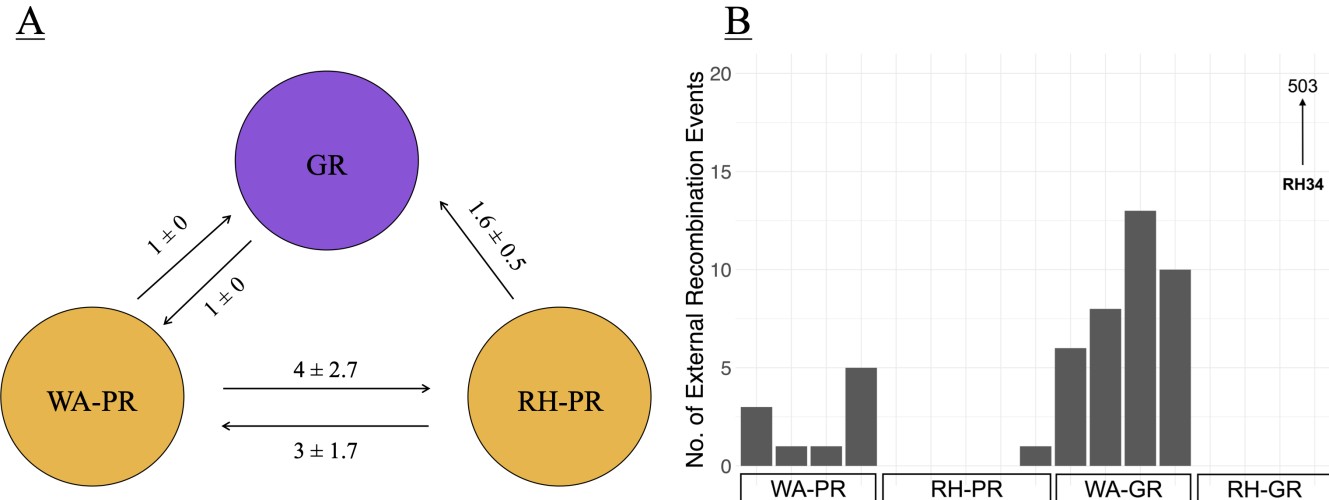

**FIG 4** Strains were grouped into three lineages based on a concatenated core gene alignment. The *griseus* clade was found to form one lineage regardless of geography, while *pratensis* strains were subdivided into two lineages, consistent with their geographic origin. Panel A shows the number of recent recombination events between the three lineages, with numbers depicting average and standard deviation of directional recent recombination events between each lineage. Panel B is a histogram of detected external recombination events for strains in each clade. RH34 had an unusually high number of recombination events and is depicted with an upward arrow as the observation value extends beyond the visible axis in this figure.

selection or paralogs, which are often associated with adaptive interactions with the environment (51). The WA-PR group saw the highest number of gene duplications including DNA segments annotated as mobile genetic elements like integrases and transposases.

RH34 has an unusually high number of gene duplications compared to other strains: 254 versus 14 ± 15 duplications across all other strains. One possible reason for the higher duplication rate seen in the RH34 genome could be this strain's propensity for multiple duplication events. For instance, one dimodular NRPS gene was duplicated 12 times in RH34. Some other genes were duplicated thrice, such as a gene encoding MbtH-like protein, which is known to mediate antibiotic and siderophore biosynthetic pathways in *Streptomyces coelicolor* (52).

## Secondary metabolite gene clusters are primarily vertically transmitted

Many SMGCs are found in all strains, although the total number of gene clusters in each genome is variable (Fig. 6): RH-PR has 37.4 ± 5 clusters, WA-PR 35.7 ± 1.5, RH-GR 41.7 ± 3.6, and WA-GR 55 ± 12.9. Many gene clusters are clade-restricted with strains from the same species having more SMGCs in common ($P < 0.05$, ANOVA; 50 ± 10.5 in GR, 37 ± 3.8 in PR). In contrast, only the number of lantipeptide genes varied by geography (RH > WA; $P < 0.05$).

Almost every SMGC examined was under strong negative selection pressure (Table 1). Only two loci in siderophore biosynthetic gene clusters (BGCs) showed evidence of positive selection in RH34 and WA1064 vs all other members of RH-GR (dN/dS = 2.79 and 1.32, respectively). Both sequences correspond to short ORFs upstream of siderophore synthetases.

The genome alignments of SMGCs indicate strain-level gene deletions and rearrangements, even in the presence of strong phylogenetic signal. As an example, Fig. 7 depicts a butyrolactone cluster where Segment A (demarcated by black borders) is only found in WA-GR strains. Truncated parts of Segment A are seen in the WA-PR genomes (with a rearrangement in the WA53 genome), as indicated by the black borders. Sequence annotation suggests that these shared domains are DNA-binding response regulators. The additional distinct domains shared by WA-GR strains alone correspond to SARP-family transcriptional regulators and NaeI type II restriction endonucleases. Most

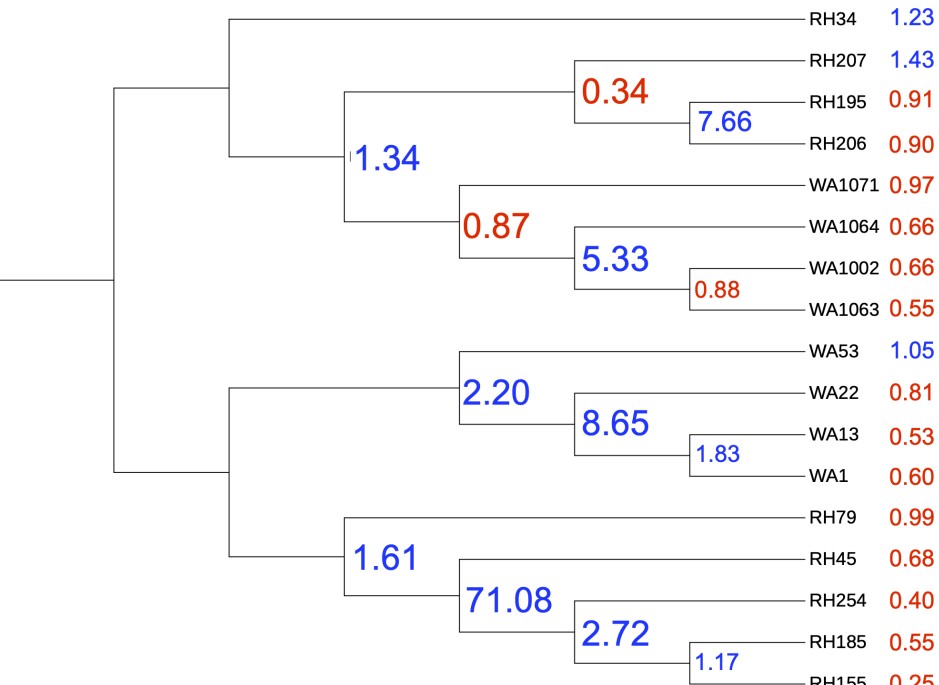

**FIG 5** The ratio of gene gain to loss is noted at each node and tip of the phylogenetic tree. Negative values (in red) indicate that more genes were lost than gained, and a positive value (blue) indicates higher gene gain compared with loss. The tips show more gene loss than the nodes, on average. Gene duplications are typically higher at nodes closer to the root of the tree (Table S3).

importantly, all WA-GR segments share a site-specific integrase seen in many *Streptomyces* genomes, indicating that these segments could have been horizontally acquired and shared with other WA *Streptomyces*.

This last finding suggested that SMGC pathways could contain multiple gene-level recombination events, resulting in phylogenetic incongruence in the pathways. To test whether vertical transmission holds true at the gene level in other SMGCs, phylogenetic trees were constructed through the maximum likelihood method, and tree topologies were then analyzed using the Shimodaira-Hasegawa (SH) test. First, homologs of genes present in all strains were used to reconstruct phylogeny for bacteriocin, lantipeptide, melanin, siderophore, and terpene gene clusters. Then, tree topologies were examined for individual, single-copy homologous genes or ORFs within each cluster. Loci in the siderophore (2/9), terpene (6/14), and melanin (1/2) gene clusters showed tree topologies inconsistent with vertical descent ($P < 0.05$, 10,000 bootstrap replicates, SH test) as compared with a maximum-likelihood core genome tree constructed using 3,110 single-copy, homologous sequences. Thus, HGT was observed even within SM gene clusters that indicated vertical descent.

## Mechanisms of gene exchange

Genomes were scanned for evidence of mobile elements, including plasmids, integrative and conjugative elements, insertion sequences, as well as gene annotations that indicated the presence of transposons, phage elements, plasmids, conjugation elements, and integrons using the approach described in Kent et al. (4).

Prophages showed the most significant difference between strains and clades out of all the MGEs we scanned. The number of phage genome fragments identified within each *Streptomyces* genome (at 100% identity) varies between strains (Fig. 8). We aligned these prophage fragments with sequences in the Actinobacteriophage database (45) to identify known actinophages in these genomes.

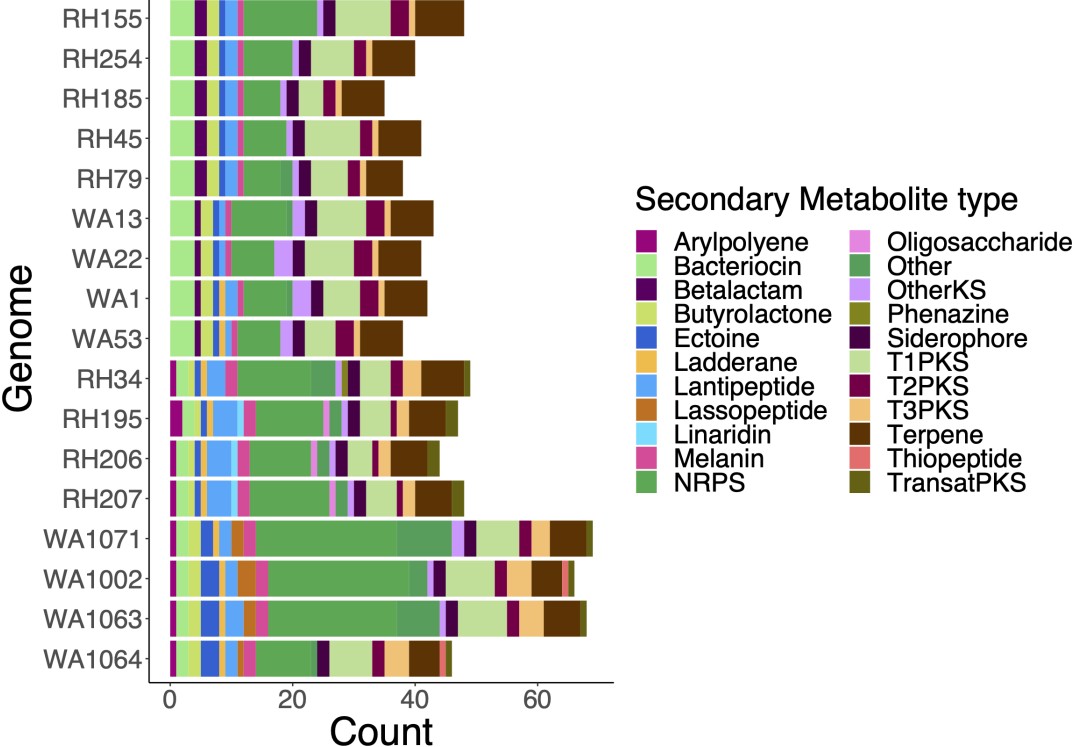

**FIG 6** The stacked bar chart shows the number of gene clusters of each secondary metabolite category per genome, and the y-axis labels are arranged in the same order as the tree in Fig. 1 to indicate the phylogenetic relationships between the genomes.

While two prophages (Beuffert and Sros11) are found in all clades, clade membership has a strong influence on patterns of host-phage interactions (two-way ANOVA, $P <$ 0.001). RH-GR strains have the highest number of prophage sequences (10.5 ± 0.6), and WA-GR had the lowest (5 ± 1.4). The WA-GR strains lack three phage signatures that are found in all other genomes (OnionKnight, SPB78, VWB) and associate with a different phage that is unique to this clade (Bing). RH-GR strains also associate with two unique phages (JackieB and pZL12). While actinophages are broadly able to infect strains in the phylum Actinomycetota, the patterns displayed by Bing, JackieB, and pZL12 suggest that phage specificity could drive clade-level HGT events via selective transduction.

## DISCUSSION

This study analyzed *Streptomyces* genomes across two species groups that were evenly distributed across phylogroup and geography. Comparative genomics methods were used to contrast gene flux across strains, with a special focus on HGT and secondary metabolite gene clusters. We predicted that phylogenetic affinity would dictate gene exchange between *Streptomyces* strains, given the species barrier to recombination, but that geography is likely to play a role in the acquisition of unique genes that may have adaptive significance to the local habitat.

Phylogenetic reconstruction and pangenome analysis show that conspecific strains are differentiated into subclades based on their isolation sites (Fig. 1 and 2). Recent recombination patterns largely followed ancestry for *griseus* strains, but recombination in *pratensis* strains was delineated by geography. Shell gene and SMGC sharing are strongly impacted by phylogeny, as are shared prophage elements. Thus, several observations indicate that shared phylogenetic history plays a dominant role in gene sharing but geography is important on a different, shorter timescale.

The geographical differentiation observed in the core genomes of these terrestrial *Streptomyces* species is in contrast to other bacteria, such as marine strains of *Vibrio*

**TABLE 1** Selection pressure (as measured by dN/dS ratio) for different SMGCs shared by all strains in a clade or the data set[a]

| SMGC type | Clade | dN/dS range |
|---|---|---|
| Arylpolyene | GR | 0.001–0.138 |
| Bacteriocin | All | 0.001–0.553 |
| Betalactams | PR | 0.001–0.072 |
| Ectoine | All | 0.001–0.495 |
| Lantipeptides | All | 0.001–0.551 |
| Melanin | All | 0.001–0.588 |
| Siderophore | All | 0.035–2.795 |
| Terpene | All | 0.001–0.797 |
| Butyrolactone | All | 0.001–0.164 |
| Lassopeptides | WA-GR | 0.001–0.252 |
| Thiopeptides | WA-GR | 0.001–0.22 |
| NRPS | All | 0.001–0.846 |
| PKS | All | 0.001–0.846 |

[a]Presence of a specific clade in the second column indicates presence of a clade-specific SMGC, not that selection pressure exists between those strains.

*cholerae*, which exhibit little biogeographic structure due to high dispersal rates (11). *Streptomyces* have previously been identified to be dispersal limited at regional and local scales (54, 55), and it is possible that the strains in this study experienced non-equilibrium evolutionary dynamics associated with population expansion and subsequent dispersal limitation, which could contribute to reduced gene flow and the geographical structure seen in Fig. 1. Even within the same phylogroup, *griseus* strains display higher gene flow, leading to the genetic cohesiveness seen in Fig. 4A. This finding is supported by the ANI values in Table S2, where the average ANI between *griseus* strains is higher when compared with the *pratensis* strains (96.3 ± 2.9).

Intraspecies recombination is more common than interspecies recombination for the two species in this study; GR and PR strains share 13 recent recombination events as compared with 29 recent and 457 ancient events within PR. GR-GR recombination events are more common than within-PR recombination, and the geographical split seen in the *pratensis* lineage could serve as an indicator of how isolation-by-distance might increase genetic diversity in bacterial populations and eventually lead to sister lineages diverging beyond the species boundary. However, interspecies HGT events are not uncommon, as demonstrated by the roughly 3,000 potential external recombination events identified by HGTector.

While shell gene content varied by phylogeny, strain-specific (unique) gene content varied by site. Given that unique genes, by definition, are only present in one genome and thus have not yet swept through the population, strain-specific genes might be indicative of regular samplings of the local gene pool in a given habitat. Changes in core and shell genes are likely integrated over much longer periods of evolutionary and geological history and thus reflective of conserved phylogenetic inheritance. Given that strains isolated from Washington have larger genomes, a higher number of unique genes and external recombination events, we propose that both *pratensis* and *griseus* strains expanded to Washington from Wisconsin in the recent past. Previous observations have indicated that southern Wisconsin (where Rhinelander is located) was part of a glacial refugium during the Quaternary (56), thus making it one of the earlier sites available for colonization once temperatures rose. Relaxed selection often follows the removal of a population bottleneck when a founder population emigrates to an unoccupied habitat, resulting in a temporary phase of genome expansion by gene acquisition (57). This idea is supported by Fig. 5, which shows high rates of gene loss at the tips of the tree. This pattern suggests the possibility of population expansion until more recent times, when gene loss becomes the dominant process that drives changes in genome size (e.g., between RH-GR and WA-GR or within RH-PR). Earlier studies have suggested that gene loss can be used by *Bradyrhizobium* strains to reduce genome size when exposed to

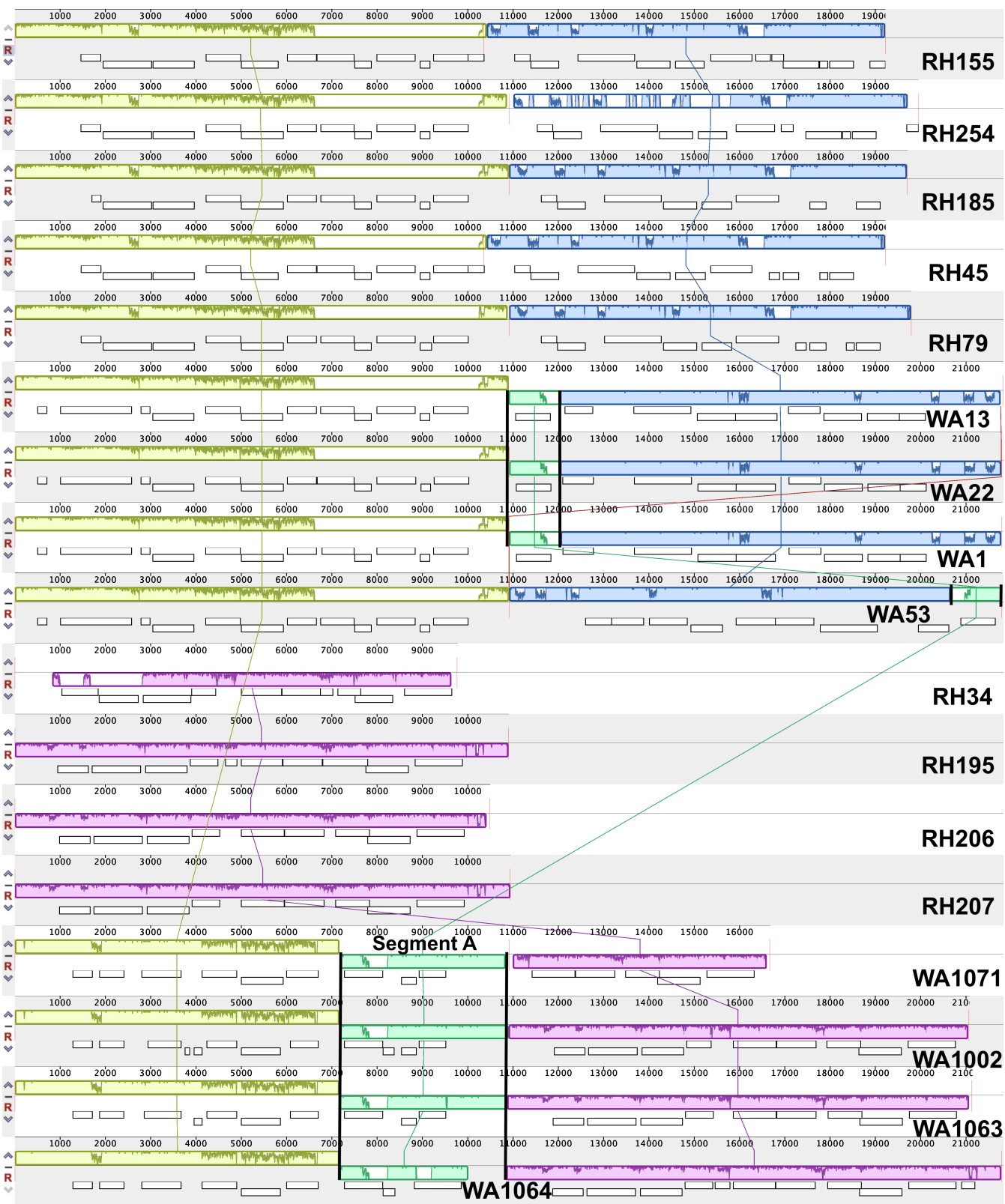

**FIG 7** Multiple genome alignment of the butyrolactone-encoding gene clusters indicates regions of phylogenetic (lime green, blue, and pink local collinear blocks or LCBs) and geographic specificity (mint green LCB in WA genomes). The genomes are arranged in the same order as Fig. 1, and each panel contains regions of similarity (LCBs of different colors and % similarity plots within each block) connected by lines of the same color across genomes, overlaid with the individual contigs associated with this SMGC. Numbers on the x-axis indicate position on the alignment. Genome alignment and visualization were performed using progressiveMauve (53).

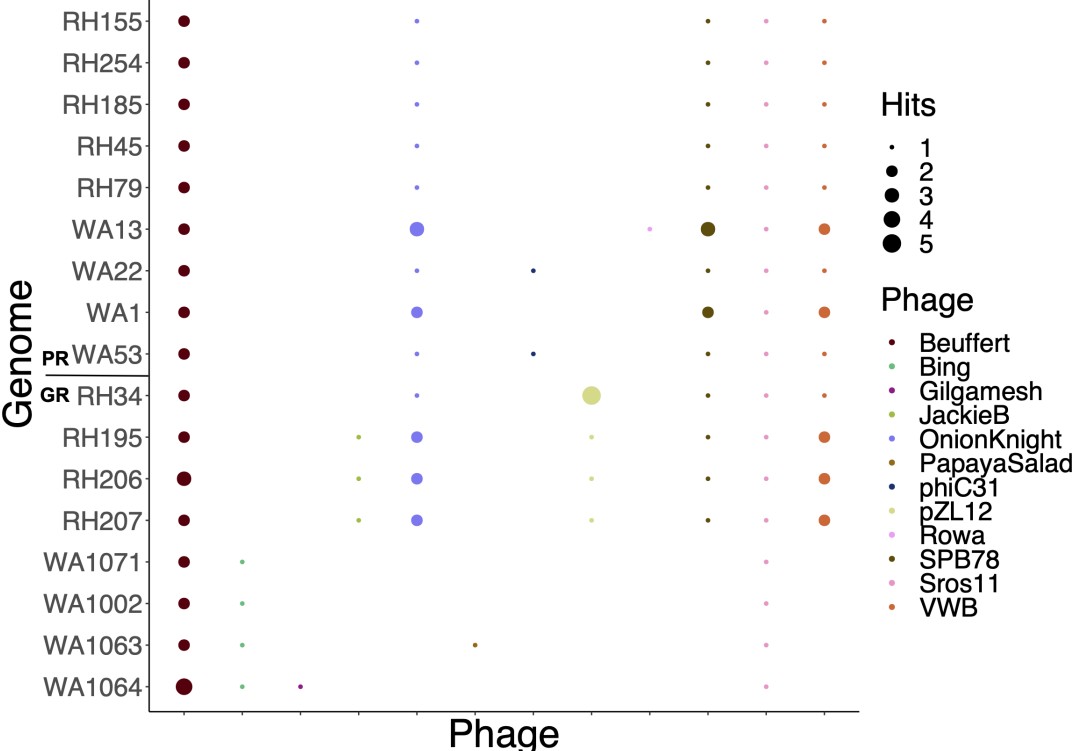

**FIG 8** Prophage sequences were distributed unevenly across all *Streptomyces* genomes. Bubble colors represent the phage sequence aligned to the *Streptomyces* strain on the y-axis, and the size of the bubble represents the number of unique hits from the phage. The WA-GR clade in particular showed a distinct phage element signature. Sequences from two phage genomes (Beuffert and Sros11) were present at 100% identity in all 17 strains.

environmental stressors (58), and it is possible that genome size reduction may reflect the end of the relaxed selection period but also indicate increasing genome efficiency under changing ecological conditions.

Although the accessory genes in this study show localization by both phylogeny and geography, gene gain and loss patterns indicate regular acquisition and loss of genes encoding amino acid and vitamin biosynthesis for all strains. Our knowledge of vitamin and amino acid auxotrophy is still meager, but increasing evidence suggests that nutrient sharing in microbial communities is an important and complex process that shapes their ecology and evolution (59). Future studies on soil *Streptomyces* should focus on how gain and loss of the capacity to biosynthesize specific amino acids and vitamins (particularly the B vitamin family) could provide an advantage in adapting to new habitats.

We expected that secondary metabolite gene clusters would follow site-specific distribution patterns, given their potential for adaptation to local environmental conditions and interactions with neighboring strains. Fig. 6 indicates that many SMGCs were derived from a common ancestor of both *S. griseus* and *pratensis* and hence present in all strains. While there are species-specific SMGCs and even some clade-specific gene clusters, there are no SMGCs unique to either site. This finding is in alignment with recent studies (60) that have found lineage specificity of SMGCs in other bacteria, which supports the idea that secondary metabolites could be involved in some species' abilities to tolerate environmental stressors or compete for resources.

However, phylogenetic incongruence in three SMGCs indicates that HGT has likely occurred at the individual gene level despite the overall pattern of vertical inheritance, which is in agreement with the lateral gene transfer patterns noted by (61). Ziemert et al. (62) have suggested that SMGCs in *Salinispora* (a closely related Actinomycete) represent a shared population resource, and that gene cluster mobility and rearrangement within

conspecific strains can be used to enhance fitness and niche utilization. The *Streptomyces* strains in this study display phylogenetic conservation of BGC pathways with evidence for horizontal transfer of gene segments within some pathways and could thus be adopting a similar strategy to optimize fitness. That being said, we recognize that short read sequencing has well-known limitations with respect to genome assembly, particularly with poor coverage of repetitive regions (63) like those found in some MGEs. Future work in this area would ideally rely on hybrid genome assembly that would supplement the coverage provided by Illumina sequencing with long-read technology that could improve coverage of low-resolution genomic fragments and close gaps in genome assemblies (64).

The mechanisms of HGT in environmental microbes are hard to identify. This study confirms the importance of conjugation and transduction for HGT in soil *Streptomyces*. Plasmids and actinophages in particular could be important drivers of HGT. While plasmids seem to be environmentally acquired, phage associations show species-level specificity. Many bacteriophages carry accessory genes that serve adaptive functions and are under purifying selection in bacterial genomes. In some enteric bacteria, genomes are thought to be in constant contact with closely related phages, with phage-derived genes frequently replaced by other incoming prophage-derived genes (65). We are still discovering the host specificity and prevalence of actinophages in the soil, but this study suggests that phage specificity could be important in constraining DNA transfer within closely related populations at the same location.

A better understanding of HGT dynamics in the environment will ultimately lead to a deeper understanding of bacterial speciation and taxonomic delineations, particularly in taxa with high rates of HGT and cosmopolitan lifestyles like *Streptomyces*. A previous analysis has suggested that the traditional depiction of evolution using phylogenetic trees should be reconsidered for such species (66). A reticulate model may be a better representation of evolution for species with high recombination rates, wherein recombination or hybridization events are depicted as additional edges that essentially depict evolutionary history as a network rather than a bifurcating tree (67).

Future work in this area should focus on how HGT varies with different abiotic factors and across gradual shifts in microhabitats. In particular, experimental designs should account for the different temporal scales at which phylogenetic and geographical barriers to gene exchange may operate. Analysis of accessory gene content and pangenome variation will illuminate the contribution of HGT to bacterial adaptation, niche partitioning, and ecotype differentiation in soil. Finally, experiments that identify the diversity of phages in the local community, and plasmids or other conjugative elements in the local gene pool will improve our mechanistic understanding of HGT. Horizontal gene transfer is clearly relevant at multiple temporal and spatial scales and contributes to the richness of bacterial species and lifestyles in the environment. It is a vital ecological and evolutionary force that can create differentiation and cohesion between lineages.

## ACKNOWLEDGMENTS

This work was supported by the National Science Foundation under Award No. 1456821 to D.H.B. and Award No. 2055120 to C.P.A. The funders had no role in study design, data collection and analysis, decision to publish, and preparation of the manuscript. The findings in this study do not necessarily reflect the views and policies of the authors' institutions and funders.

## AUTHOR AFFILIATIONS

[1]School of Integrative Plant Science, Cornell University, Ithaca, New York, USA
[2]Department of Microbiology, Cornell University, Ithaca, New York, USA

## PRESENT ADDRESS

Janani Hariharan, Department of Biology, Bucknell University, Lewisburg, Pennsylvania, USA

Cheryl P. Andam, Department of Biological Sciences, University at Albany, State University of New York, Albany, New York, USA

## AUTHOR ORCIDs

Janani Hariharan  http://orcid.org/0000-0002-5535-4119
Cheryl P. Andam  http://orcid.org/0000-0003-4428-0924

## FUNDING

| Funder | Grant(s) | Author(s) |
| --- | --- | --- |
| National Science Foundation | 1456821 | Janani Hariharan |
| | | Daniel H. Buckley |
| National Science Foundation | 2055120 | Cheryl P. Andam |

## AUTHOR CONTRIBUTIONS

Janani Hariharan, Data curation, Formal analysis, Investigation, Methodology, Visualization, Writing – original draft, Writing – review and editing | Cheryl P. Andam, Conceptualization, Data curation, Formal analysis, Funding acquisition, Investigation, Visualization, Writing – review and editing | Daniel H. Buckley, Conceptualization, Project administration, Resources, Supervision, Writing – review and editing

## DATA AVAILABILITY

Genome assemblies used in this study can be obtained from NCBI BioProjects PRJNA401484 and PRJNA934405.

## ADDITIONAL FILES

The following material is available online.

### Supplemental Material

**Supplemental material (Spectrum02958-25-s0001.docx).** Tables S1 to S3.

### Open Peer Review

**PEER REVIEW HISTORY (review-history.pdf).** An accounting of the reviewer comments and feedback.

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
