## [Reviewer comments · Microbiology Spectrum]

Microbiology Spectrum

Biogeographical and phylogenetic constraints on horizontal gene transfer and genome evolution in *Streptomyces*

Janani Hariharan, Cheryl Andam, and Daniel Buckley

Corresponding Author(s): Janani Hariharan, Cornell University

Review Timeline:

Submission Date:	September 16, 2025
Editorial Decision:	October 22, 2025
Revision Received:	November 11, 2025
Accepted:	November 21, 2025

Editor: Blaire Steven

Reviewer(s): Disclosure of reviewer identity is with reference to reviewer comments included in decision letter(s). The following individuals involved in review of your submission have agreed to reveal their identity: Sébastien Bontemps-Gallo (Reviewer #1)

Transaction Report:

DOI: <https://doi.org/10.1128/spectrum.02958-25>

Re: Spectrum02958-25 (Biogeographical and phylogenetic constraints on horizontal gene transfer and genome evolution in *Streptomyces*)

Dear Dr. Janani Hariharan:

Thank you for the privilege of reviewing your work. Below you will find my comments, instructions from the Spectrum editorial office, and the reviewer comments.

After having read the manuscript and receiving reviewer comments I am happy to report that this manuscript should be ready for publication after relatively minor revisions.

Revision Guidelines

Sincerely,
Blair Steven
Editor
Microbiology Spectrum

Reviewer #1 (Comments for the Author):

Hariharan et al. study dissects how phylogeny versus geography shape horizontal gene transfer (HGT) and genome evolution in *Streptomyces* by analyzing 17 isolates from two sites spanning the *S. pratensis* and *S. griseus* phylogroups. Leveraging ANI-based relatedness, pangenome profiling, and recombination inference, the authors show that core-genome homologous

recombination is predominantly constrained by shared ancestry: they detect 1014 events (557 recent/457 ancestral) with intra-species exchange greatly exceeding inter-species exchange across sites. In contrast, the accessory "shell" genome exhibits a local gene pool effect, although shell content overall tracks phylogeny, significantly more shell genes are shared by different species that co-occur at the same site than by conspecifics across sites, indicating geography can channel gene flow at shorter evolutionary scales. Secondary-metabolite gene clusters are abundant (about 35-55 per genome, clade-dependent), and around 60% occur in all strains, consistent with vertical inheritance from a common ancestor. Nevertheless, gene-level phylogenetic incongruence within siderophore, terpene, and melanin clusters reveals mosaic HGT within otherwise vertically maintained pathways. Finally, prophage complements vary by clade and site, suggesting that phage specificity could mediate selective transduction and contribute to observed patterns of gene exchange. Together, these results support a two-scale model: core-genome exchange follows lineage history (phylogenetic constraint), whereas parts of the accessory genome (especially shell genes) reflect geographic co-occurrence and local mobile genetic elements shaping rapid, environment-linked genome dynamics.

Major

1/ With n=17 across only two sites and two phylogroups, several key inferences may be sensitive to limited spatial replication and lineage coverage. A risk-of-error analysis for recombination-directionality estimates would strengthen conclusions.

2/ Draft genomes from GS FLX-based libraries can fragment repetitive/mobile regions. Some inferences about MGEs, recombination tracts, and SMGC structure may be under- or mis-resolved without long reads (PacBio, Nanopore) or hybrid assemblies. Please discuss biases and, if feasible, validate key regions with long-read data.

3/ The manuscript emphasizes vertical transmission of SMGCs, yet shows gene-level incongruence in siderophore/terpene/melanin clusters. Could you quantify the frequency of these incongruences relative to a null model, and clarify how this affects the "mostly vertical" conclusion.

Minor

1/ Clarify how ANI thresholds were applied to define species boundaries and handle borderline cases.

2/ In the methods, provide versions/parameters for Roary, OrthoFinder, fastGEAR, ClonalFrameML, HyPhy aBSREL, PAML models used per analysis.

We thank the anonymous reviewer for their suggestions. Our replies are in blue below.

Major

1/ With $n=17$ across only two sites and two phylogroups, several key inferences may be sensitive to limited spatial replication and lineage coverage. A risk-of-error analysis for recombination-directionality estimates would strengthen conclusions.

Thank you for the suggestion. The software we used to detect recombination events, i.e. fastGEAR, performs a step that removes false-positives (Type I errors). Here is the relevant description from the publication that describes fastGEAR: “*To prune these false positive findings, we monitor the locations of SNPs between the target strain and its ancestral lineage (for recent recombinations) or between the two lineages (for ancestral recombinations) within and between the claimed recombinant segments. We apply a simple binomial test to compute a Bayes factor (BF; see, e.g., Bernardo and Smith, 2001), that measures how strongly the changes in SNP density support a recombination, and we use a threshold $BF = 1$ for recent recombinations and $BF = 10$ for ancestral recombinations for additional pruning of recombinations proposed by the HMM analyses. These thresholds represent a compromise between false positive rate and power to detect recombinations. Recombinations with BFs less than the threshold are not reported at all, and the estimated BFs for the remaining recombinations are included in the output.*”

From Mostowy R, Croucher NJ, Andam CP, Corander J, Hanage WP, Marttinen P. 2017. Efficient Inference of Recent and Ancestral Recombination within Bacterial Populations. *Mol Biol Evol* 34(5):1167-82. <https://doi.org/10.1093/molbev/msx066>

For the recombination estimates in our paper, we quantify the variation in detected recombination events in Fig. 4a. In addition, we have added an effect size estimate for the contribution of phylogeny and geography for the fastGEAR-detected recombination effects at line 289.

2/ Draft genomes from GS FLX-based libraries can fragment repetitive/mobile regions. Some inferences about MGEs, recombination tracts, and SMGC structure may be under- or mis-resolved without long reads (PacBio, Nanopore) or hybrid assemblies. Please discuss biases and, if feasible, validate key regions with long-read data.

We agree with this comment. Lines 485-490 have been updated with a discussion of the biases and a recommendation to use hybrid assembly methods for future investigations of HGT using comparative genomics.

3/ The manuscript emphasizes vertical transmission of SMGCs, yet shows gene-level incongruence in siderophore/terpene/melanin clusters. Could you quantify the frequency of these incongruences relative to a null model, and clarify how this affects the "mostly vertical" conclusion.

We used the Shimodaira-Hasegawa test to detect phylogenetic incongruence across gene loci in each SMGC (line 369). This test uses the likelihood ratio test statistic, and generates a null distribution by resampling all candidate trees. We ran the SH test with 10,000 bootstraps wherein bootstrap replicates are used to generate the null statistic for each tree.

Generation of the null distribution is followed by calculating a p-value for each tree to ascertain if its difference from the maximum likelihood tree's score (in our case, a core genome tree generated using an

alignment containing 3,110 homologous protein sequences) can be explained under the null hypothesis. We report this result in the same paragraph (line 373). Thus, we have already compared the phylogenetic incongruence at each locus to a null distribution. Our inference of HGT within predominantly vertically-transferred gene clusters is based on these results.

We have added more details about the null model assumptions of the test in the Methods section, and specified how the core genome tree was constructed.

Minor

1/ Clarify how ANI thresholds were applied to define species boundaries and handle borderline cases.

This information has been added to lines 151-154.

2/ In the methods, provide versions/parameters for Roary, OrthoFinder, fastGEAR, ClonalFrameML, HyPhy aBSREL, PAML models used per analysis.

Thank you; this information has been added for all tools except fastGEAR, which only has one version. In addition, we have updated the following software/tools with version numbers: prokka, fastANI, mugsy, MUSCLE, and Notung.

We used HyPhy v2.5.36 for an analysis that was removed from the manuscript prior to submission. This sentence has now been deleted since those results are no longer reported in the manuscript.

Re: Spectrum02958-25R1 (Biogeographical and phylogenetic constraints on horizontal gene transfer and genome evolution in *Streptomyces*)

Dear Dr. Janani Hariharan:

After having read the response to reviewers comments, I am happy to report I am recommending the article for publication.

Your manuscript has been accepted, and I am forwarding it to the ASM production staff for publication. Your paper will first be checked to make sure all elements meet the technical requirements. ASM staff will contact you if anything needs to be revised before copyediting and production can begin. Otherwise, you will be notified when your proofs are ready to be viewed.

Sincerely,
Blair Steven
Editor
Microbiology Spectrum